# Readiness, barriers, and attitude of students towards online medical education amidst COVID-19 pandemic: A study among medical students of Ebonyi State University Abakaliki, Nigeria

**Edmund Ndudi Ossai[1,2], Irene Ifeyinwa Eze[1,2]\*, Chukwuma David Umeokonkwo[2], Chukwuemeka Obioma Izuagba[1], Lawrence Ulu Ogbonnaya[1,2]**

1 Department of Community Medicine, Ebonyi State University Abakaliki, Abakaliki, Nigeria, 2 Department of Community Medicine, Alex Ekwueme Federal University Teaching Hospital Abakaliki, Abakaliki, Nigeria

\* jorenebiz@yahoo.com

## Abstract

### Introduction

The COVID-19 pandemic caused massive disruption to medical education in Nigeria, necessitating the call for online medical education in the country. This study assessed the readiness, barriers, and attitude of medical students of Ebonyi State University Abakaliki, Nigeria, to online medical education.

### Methods

A cross-sectional study design was employed. All matriculated medical students of the university participated in the study. Information was obtained using a pre-tested, semi-structured questionnaire which was self-administered. Good attitude towards information and communication technology (ICT) based medical education was determined by the proportion of respondents correctly answering 60% of nine variables. Readiness for online classes was determined by the proportion of students who preferred either a combination of physical and online lectures or only online medical education amidst the COVID-19 pandemic. Chi-square test and multivariate analysis using binary logistic regression analysis were used in the study. A p-value of <0.05 determined the level of statistical significance.

### Results

Four hundred and forty-three students participated in the study (response rate; 73.3%). The mean age of the students was 23.0±3.2 years. The majority of the respondents, 52.4%, were males. The students' most preferred sources for studying before the COVID-19 pandemic included textbooks, 55.1% and lecture notes, 19.0%. The commonly visited websites included Google, 75.2%, WhatsApp, 70.0% and YouTube, 59.1%. Less than half, 41.1%, have a functional laptop. The majority, 96.4%, have a functioning email address, while

**Data Availability Statement:** All relevant data are within the paper and its Supporting information files.

**Funding:** The author(s) received no specific funding for this work.

**Competing interests:** The authors have declared that no competing interests exist.

33.2% participated in a webinar during the COVID-19 pandemic. Though 59.2% had a good attitude towards online medical education, only 56.0% expressed readiness for online medical education. The major barriers to online medical education included poor internet connectivity, 27.1%, poor e-learning infrastructure, 12.9% and students not having laptops, 8.6%. Predictors of readiness for online medical education included previous participation in a webinar, AOR = 2.1, (95%CI: 1.3–3.2) and having a good attitude towards IT-based medical education, AOR = 3.5, (95%CI: 2.3–5.2).

## Conclusions

The majority of the students showed readiness for online medical education. Lessons from COVID-19 pandemic necessitate the initiation of online medical education. University authorities should ensure that every enrolled medical student owns or have access to a dedicated laptop through a university-mediated arrangement. Adequate attention should be given to the development of e-learning infrastructure, including steady internet services within the confines of the university.

## Introduction

Worldwide, the COVID-19 pandemic has affected all sectors, including the educational sector [1]. The pandemic created an enormous disruption of education systems in history, involving nearly 1.6 billion learners in more than 190 countries and continents [2], adversely affecting students' academic expectations in the sub-Saharan Africa [3–5]. The fast spread of COVID-19 brought about many changes in the educational sector, including restrictions on movement and lockdowns of institutions of learning [6, 7]. In Nigeria, the COVID-19 pandemic impacted higher institutions following the lockdown in many ways: disruption of the academic calendar, teaching and learning gap, loss of workforce, reduction of international education, and cancellation of local and international conferences [8]. Furthermore, the coronavirus pandemic exposed massive socio-economic inequality and learning disparity in Nigeria's education system, with wealthy families sending their children to private schools equipped for remote learning in contrast to the poor public school facilities [9].

The educational landscape is undergoing a radical transformation to cushion the effect of the pandemic and revamp the education system, including shifting toward the e-learning [10, 11]. E-learning can enhance knowledge efficiency through easy access to a large amount of information within the global village, promoting personal knowledge accumulation and group knowledge sharing [12, 13]. Furthermore, e-learning increases satisfaction and decreases stress by enabling students and lecturers to participate in class in their comfort zone and is cost-effective as it reduces travel time and infrastructural development in terms of buildings [14, 15].

Even with the numerous benefits of e-learning, developing countries, including Nigeria, are still challenged to shift from the traditional teaching method to e-learning during the pandemic [16]. The challenges arise from varying degrees of preparedness of the institutions, including the inadequate skill of lecturers [16], social inequalities [1] and lack of infrastructures, irregular power supply, high cost of internet data services and personal computer (PC) and Laptop, poor internet connectivity [17], all of which threaten its applicability. Furthermore, although around 83% of Nigerians have mobile phone connections, the proportion is

skewed towards the few high socio-economic and urban households, most of whom are private school students with a learning advantage over their public school peers [9]. Therefore, assessing the barriers to e-learning in terms of the availability of digital facilities, including the adequacy in quantity and quality, can help academic administrators create mechanisms that can enhance users' attitudes and readiness for adopting the e-learning platform.

E-learning system has become the most preferred learning platform during global pandemic periods such as COVID-19, where movement is restricted [10, 11]. However, there needs to be a more empirical examination of the factors underlying the adoption of e-learning systems. Factors such as perceived usefulness of e-learning, ease of use, pressure to use, good technical skills and the availability of resources needed to use e-learning impact students' attitudes and readiness towards the e-learning [18, 19]. Yet, assessment of the user characteristics in e-learning systems, especially in a developing country like Nigeria, appears to be limited, despite being a prerequisite to introducing successful e-learning systems [18]. Successful system implementation and adoption by learners requires a solid understanding of user acceptance processes with these technologies. Assessing the attitude is vital in analysing consumer readiness to adopt, as a favourable attitude shows a greater probability that learners will be ready to accept the new learning system [19, 20]. Acceptance of the new learning system will be worthwhile as earlier studies reported enhanced knowledge efficiency [12, 13] and mean performance of users in favour of online education compared to the same course delivered face to face [21]. Since users' views tend to impact the advancement of the e-learning [18, 19], understanding how students perceive and react to elements of e-learning and factors that influence readiness for effective e-learning is essential and timely. This study aimed to ascertain the barriers, attitudes, and readiness for adopting e-learning among medical students in a Nigerian university.

## Methods

### Description of the study area

The College of Health Sciences of Ebonyi State University Abakaliki, Nigeria, was established in 1999, the same year the university was established. Based on the recommendation of the Nigerian Medical and Dental Council, the university admits an average of 100 medical students every year. The study of Medicine in Nigeria runs for six years, each year being recognized as a level. The first year of study (also called 100 level) is called the preliminary year, and during this period, academic activities take place at the Faculty of Science of the University. The second and third years of study are the pre-clinical training period, while the fourth to sixth year is the clinical training period. The university's medical students receive clinical training at Alex Ekwueme Federal University Teaching Hospital Abakaliki, Nigeria.

**Impact of COVID-19 pandemic on medical education.** The university, including the medical school, was closed down by the Nigeria Government during the peak of the COVID-19 pandemic on 19th March 2020; after the World Health Organization declared the COVID-19 outbreak a global pandemic on 11th March 2020. The closure involved all higher institutions in the country and aimed to curtail the spread of the virus. Like many public universities in Nigeria, the university did not initiate any form of online learning during the period the university was closed down. The study occurred between September and October 2021, after the university was re-opened in February 2021 for academic activities following the COVID-19 lockdown.

### Study design and study population

A cross-sectional design was employed. The Strengthening the Reporting of Observational Studies in Epidemiology (STROBE) was used to construct and conduct the study. All

matriculated medical students at the university were included in the study. The students who were not present on the days of data collection and those who refused to consent to participate were excluded from the study. A total of four hundred and forty-three (443) medical students participated in the study representing a response rate of 73.8%.

## Study instrument

A pre-tested, semi-structured, self-administered questionnaire that the researchers designed after a review of some literature [8, 13] was used for data collection. The questionnaire was pretested on 45 medical students (10% of the study population) at another university.

## Data management

Data entry and analysis were done using International Business Machine (IBM) Statistical Package for Social Sciences (SPSS), version 25. Chi-square test and multivariate analysis using binary logistic regression analysis were used in the research, and a p-value of <0.05 determined the level of statistical significance. The outcome measure of the study was the readiness of the medical students to commence online medical education at the university. This was represented by the proportion of students who preferred in-class and e-learning approaches and e-learning methods only in response to how to continue medical education amidst the COVID-19 pandemic. Furthermore, a good attitude towards information and communication technology (ICT) based medical education was determined by the proportion of respondents correctly answering 60% of nine variables. In determining the predictors of the readiness of the students to commence online medical education, variables with a p-value of <0.2 on bivariate analysis were included in the binary logistic regression model. The binary logistic regression analysis results were presented using an adjusted odds ratio (AOR) and 95% confidence interval. A p-value of <0.05 determined the statistical significance level.

## Ethical consideration

Ethical approval for the study was obtained from the Research and Ethics Committee of Ebonyi State University Abakaliki, Nigeria. (Reference number, EBSU/DRIC/UREC/Vol.04/ 064). The students signed a written informed consent form before participating in the study. The nature of the research and their involvement were known to them. The students were informed that participation in the survey was voluntary and assured that all information provided through the questionnaire would be kept confidential.

## Results

Table 1 shows the socio-demographic characteristics of the respondents. Four hundred and forty-three students participated in the study (response rate; 73.3%). The mean age of the students was 23.0±3.2 years. The majority of the respondents, 52.4%, were males and single 428 (96.9%). Most students were in 300 academic level 102(23.0%) and 400 academic level 103 (23.3%).

Table 2 shows access to and application of internet services among the respondents. Majority of the respondent had an android phone 373 (84.2%2), a functional e-mail address 427 (96.4%) and access to the internet 414 (93.5%). Less than half, 184 (41.1%), had a functional laptop, and 147 (33.25) participated in webinars during the pandemic. The most preferred source for studying before the COVID-19 pandemic were textbooks 224(55.1%) and lecture notes 84(19.0%). The commonly visited websites included google 333 (75.2%), WhatsApp 310

**Table 1. Socio-demographic characteristics of respondents.**

| Variable | (n = 443) Frequency (%) |
|---|---|
| **Age of respondents** | |
| Mean±(SD) | 23.0±3.2 |
| **Age of respondents in groups** | |
| <20 years | 37 (8.4) |
| 20–24 years | 291 (65.7) |
| 25–29 years | 99 (22.3) |
| ≥30 years | 16 (3.6) |
| **Gender** | |
| Male | 232 (52.4) |
| Female | 211(47.6) |
| **Academic level** | |
| 100 level | 23 (5.2) |
| 200 level | 93 (21.0) |
| 300 level | 102 (23.0) |
| 400 level | 103 (23.3) |
| 500 level | 64 (14.4) |
| 600 level | 58 (13.1) |
| **Marital status** | |
| Single | 428 (96.6) |
| Married | 15 (3.4) |
| **Religion of respondent** | |
| Christianity | 437 (98.6) |
| Islam | 5 (1.2) |
| Traditional religion | 1 (0.2) |
| **Ethnicity of respondent** | |
| Igbo | 424 (95.7) |
| Others** | 19 (4.3) |
| **Educational attainment of Father** | |
| No formal education | 28 (6.3) |
| Primary education | 48 (10.8) |
| Secondary education | 82 (18.5) |
| Tertiary education | 285 (64.3) |
| **Educational attainment of Mother** | |
| No formal education | 28 (6.3) |
| Primary education | 45 (10.2) |
| Secondary education | 77 (17.4) |
| Tertiary education | 293 (66.1) |

(70%), and YouTube 262 (59.1%). School-based activities that the internet was mostly used were course registration 362 (81.7%) and payment of school fees 327(73.8%).

Table 3 shows readiness to online medical education. Majority of the respondents, 235 (53.3%), expressed readiness for a combination of in-class and e-learning approaches to medical education amidst the Covid-19 pandemic. Only 12 (2.7%) preferred e-learning methods only.

Table 4 shows barriers to online medical education. The major barriers to online medical education included poor internet connectivity, 120 (27.1%), poor e-learning infrastructure, 57

**Table 2. Access to and application of internet services among the respondents.**

| Variable | (n = 443) Frequency (%) |
| --- | --- |
| **Respondent has a functional laptop** | |
| Yes | 182 (41.1) |
| **Ownership of devices** | |
| Android phone | 373 (84.2) |
| Tablet | 49 (11.1) |
| iphone | 46 (10.4) |
| Internet modem | 46 (10.4) |
| **Respondent has a functional email address** | |
| Yes | 427 (96.4) |
| **Respondent has access to the internet** | |
| Yes | 414 (93.5) |
| **Respondent participated in Webinar during the pandemic** | |
| Yes | 147 (33.2) |
| **The most preferred source for studying in medical school before COVID-19**\*\* | |
| Textbooks | 244 (55.1) |
| Lecture notes | 84 (19.0) |
| E-textbooks | 65 (14.7) |
| Internet | 39 (8.8) |
| Uncertain | 11 (2.5) |
| **Commonly visited websites**\*\* | |
| Google | 333 (75.2) |
| WhatsApp | 310 (70.0) |
| YouTube | 262 (59.1) |
| Facebooks | 164 (37.0) |
| Instagram | 127 (28.7) |
| Twitter | 114 (25.7) |
| Google scholar | 40 (9.0) |
| Yahoo | 39 (8.8) |
| **School-based activities that use the Internet before COVID-19**\*\* | |
| Course registration | 362 (81.7) |
| Payment of school fees | 327 (73.8) |
| Checking results after examination | 301 (67.9) |
| Further reading after lectures | 242 (54.6) |
| Research work/projects | 227 (51.2) |
| Registration for examinations | 224 (50.6) |
| Submission of assignments | 123 (27.8) |
| Lectures | 36 (8.1) |

\*\*multiple responses encouraged

(12.9%), and students not having laptops, 38(8.6%). The majority of the respondents, 275 (62.1%), reported that the university has no functional virtual library.

Table 5 shows the attitude towards information and technology-based medical education. Overall good attitude towards online medical education was noted in 263 (59.4%) respondents. However, 68.8% of the respondents viewed that lectures for medical teaching should be made available on the university e-learning portal, and 60.9% opined that online learning platforms should complement in-class teachings in medical school.

**Table 3. Readiness for online medical education among the respondents.**

| Variable | (n = 443) Frequency |
|---|---|
| **Approach to medical education amidst the COVID-19 pandemic** | |
| Physical learning only | 184 (41.5) |
| In-class and e-learning approaches | 235 (53.3) |
| Rely on e-learning methods only | 12 (2.7) |
| Uncertain | 11 (2.5) |
| **Readiness for online medical education** | |
| Yes | 248 (56.0) |
| No | 195 (44.0) |

Table 6 shows predictors of readiness for online medical education. The predictors of readiness for online medical education included previous participation in a webinar, AOR = 2.1, (95%CI: 1.3–3.2) and having a good attitude towards ICT-based medical education, AOR = 3.5, (95%CI: 2.3–5.2).

## Discussions

The study found that students' major barriers to online medical education included poor internet connectivity, poor e-learning infrastructure, unavailability of personal laptops, and lack of

**Table 4. Barriers for online medical education among the respondents.**

| Variable | (n = 443) Frequency (%) |
|---|---|
| **Approach to medical education amidst the COVID-19 pandemic** | |
| Physical learning only | 184 (41.5) |
| In-class and e-learning approaches | 235 (53.3) |
| Rely on e-learning methods only | 12 (2.7) |
| Uncertain | 11 (2.5) |
| **Readiness for online medical education** | |
| Yes | 248 (56.0) |
| No | 195 (44.0) |
| **The main barrier to e-learning application in medical school** | |
| Poor internet connectivity | 120 (27.1) |
| Poor e-learning infrastructure | 57 (12.9) |
| Students do not have laptops | 38 (8.6) |
| High demand on time | 36 (8.1) |
| Lack of technical skills on the part of students | 33 (7.4) |
| Cost of internet services | 31 (7.0) |
| Inadequate preparation for medical practice | 21 (4.7) |
| Reduction in the interaction between teachers and medical students | 18 (4.1) |
| Poor preparation of medical students for examinations | 13 (2.9) |
| No response | 15 (3.4) |
| **Need to increase school fees because of the introduction of e-learning** | |
| Yes | 24 (5.4) |
| No | 419 (94.6) |
| **University has a functional virtual library** | |
| Yes | 66 (14.9) |
| No | 275 (62.1) |
| Don't know | 102 (23.0) |

**Table 5. Attitude towards information & communication technology (ICT) based medical education.**

| Variable | (n = 443) Frequency (%) |
|---|---|
| Lectures for medical teaching should be made available on the university e-learning portal | 305 (68.8) |
| Physical lectures only should continually be used for medical education | 232 (52.4) |
| Every medical student should have a functional laptop | 204 (46.0) |
| Online learning platforms should be used to complement in-class teachings in medical school | 270 (60.9) |
| There is no need for the use of computers in medical education | 255 (57.6) |
| Need for training students in e-learning content development | 302 (68.2) |
| Adoption of e-learning could increase the satisfaction of medical students with training | 290 (65.5) |
| E-learning could improve the quality of learning in a medical school | 299 (67.5) |
| Need to improve e-learning infrastructure in medical school | 330 (74.5) |
| **Attitude towards ICT-based medical education** | |
| Good | 263 (59.4) |
| Poor | 180 (40.6) |

(Only correct responses indicated on the table)

functional virtual libraries in the university. This finding is consistent with the result of earlier studies on assessing the availability and utilisation of ICT resources in teaching in Nigeria, which revealed that the availability of digital resources for effective instructional delivery in schools is relatively low [22, 23]. Under such situations, even if teachers are well trained, make themselves and their instructional materials available online, and are willing to impart knowledge to the students, they are hindered by the lack or inadequacy of the technological equipment and facilities [10]. This is because, many students lack the means to access online materials from home due to poor connectivity or the unavailability of personal laptops [10]. The finding aligns with a previous study which noted that nearly half of the low-income families lacked sufficient ICT devices at home [10]. This research revealed the need for adequate availability of digital resources amidst and post covid-19 era in Nigerian Universities. This view is supported by an earlier study which showed a smoother and positive shift for schools which provided adequate resources for students and academic staff [9].

The study found that more than half of the respondents had a good attitude towards e-learning. This finding collaborates with some cross-sectional studies conducted in tertiary institutions in Nigeria, where four-fifths of the respondents had a favourable attitude and agreed that online education is the alternative measure for conversational in-class teaching and learning for future occurrences of any pandemic [19, 24]. The result is also supported by the findings of Pingle, who reported that undergraduates in India have a higher acceptance level of comfort working with computers and other e-learning packages than in the traditional face-to-face classroom [25]. As revealed from the findings, there may be a link between attitude and the perceived usefulness of the technology, as most students agreed that e-learning could improve the quality of learning and satisfaction with training. Thus, the high perceived usefulness positively influences students' attitudes towards using e-learning systems. This position is congruent with the findings of an earlier study, which suggest that the most critical belief underlying an individual's attitude towards adopting a new technology depends on the person's perceptions of the usefulness of the technology [19]. To improve students' attitudes towards e-learning, training and information sessions on e-learning need to focus primarily

**Table 6. Predictors of readiness for online medical education among the respondents.**

| Variable | Readiness for online lectures (n = 443) Frequency (%) | p-value | Adjusted odds ratio | 95%confidence interval |
|---|---|---|---|---|
| **Age of respondents in groups** | | | | |
| <20 years | 21 (56.8) | 0.761 | NA | |
| 20–24 years | 166 (57.0) | | | |
| ≥25 years | 61 (53.0) | | | |
| **Gender** | | | | |
| Male | 138 (59.5) | 0.120 | 1.307 | 0.873–1.956 |
| Female | 110 (52.1) | | 1 | |
| **Period of training** | | | | |
| Non-clinical students | 126 (57.8) | 0.448 | NA | |
| Clinical students | 122 (54.2) | | | |
| **Marital status** | | | | |
| Single | 243 (56.8) | 0.072 | 2.096 | 0.632–6.951 |
| Married | 5 (33.3) | | 1 | |
| **Have a functional laptop** | | | | |
| Yes | 107 (57.5) | 0.577 | NA | |
| No | 141 (54.9) | | | |
| **Participation in Webinar during the COVID-19 pandemic** | | | | |
| Yes | 102 (69.4) | <0.00 | 2.080 | 1.343–3.222 |
| No | 146 (49.3) | | 1 | |
| **Attitude towards ICT-based med. education** | | | | |
| Good | 181 (68.8) | <0.001 | 3.459 | 2.303–5.195 |
| Poor | 67 (37.2) | | 1 | |

ICT Information & communication technology NA Not applicable

on how the technology can help improve the efficiency, productivity, and effectiveness of students' learning process rather than on the actual use of the technology.

On students' readiness for online medical education, the study revealed that about half of the respondents expressed willingness for a combination of in-class and e-learning approaches, while a minor two percent preferred reliance on e-learning methods only. The findings agree with a study where willingness could have been better due to reported minimal online teacher-student interaction [26]. The extent of readiness may be related to the numerous barriers to online access reported by the respondents, which is common in developing countries like Nigeria.

The findings show a significant relationship between previous experience with e-learning and readiness to use an e-learning system. Those who participated in webinars during the COVID-19 pandemic were twice more ready to adopt e-learning than those who did not. The finding could be explained by the fact that users' prior experience with technology affects their views about technology in general. Users or potential adopters of the system would be ready to engage in the behaviour because of previous hands-on expertise. This assertion resonates with the student's view on training in e-learning content development. However, our findings contradict a study in Nigeria, which showed no significant relationship between the level of computer experience and intention to use e-learning systems [19]. In addition, our findings show that students with good attitudes towards using the e-learning system are three times more likely to be ready for e-learning than those with a poor attitude. Attitude indicates, to a certain

degree, the possibility of adopting certain behaviours, and students' favourable and positive attitude towards an e-learning system suggests a greater probability that they will accept it [19].

### Strengths

The main strength of this study is its focus on user characteristics which influence the adoption of e-learning systems, considering that it is a prerequisite to introducing successful e-learning systems [16]

### Study limitations

The COVID-19 pandemic adversely affected medical education in Nigeria, with the closure of all higher institutions for one year to curtail the virus's spread. However, this closure may have affected the students differently. This study took place about seven months after the medical school was re-opened. Thus, students' responses may have been influenced by the various forms affected by the closure. Also, the participation of the first-year students in the study was limited. This was due to their involvement in the university registration formalities, which were delayed because of the university's closure at the pandemic's peak. In all, the study provided insight into the readiness of medical students to commence online medical education.

## Conclusion

The study concludes that most students were ready and had a good attitude towards online medical education. However, lessons from the COVID-19 pandemic reveal lapses and gaps in adopting online medical education. These barriers to e-learning range from poor internet connectivity, poor e-learning infrastructure, unavailability of personal laptops, high internet subscription costs, and lack of functional virtual library in the university, and need to be addressed.

To improve the attitude and readiness of students for online medical education, the Nigerian government should put in place measures that ensure continuity, inclusion and equity for all learners and mitigate learning disruption during this pandemic and beyond. For example, the government should invest in providing solar-powered educational gadgets pre-loaded with offline academic resources to learners and other e-learning infrastructure, including steady internet services within the confines of the university. In addition, University authorities should ensure that every enrolled medical student owns or have access to a dedicated laptop through a university-mediated arrangement. Furthermore, the institution's authorities should implement staff training and capacity building on e-learning. To benefit from the advantages of e-learning, such as wide coverage, cost-effectiveness, uniformity, fast teaching and learning process, and rapid economic development through e-commerce, compliance with e-learning in tertiary institutions should go beyond the COVID-19 lockdown period. Thus, to facilitate students' readiness and positive attitudes and actual usage of e-learning systems, the universities should ensure the supply of suitable support in terms of technological, economic, and organizational issues aimed at guaranteeing a high level of education to their students.

## Supporting information

**S1 Dataset.**
(SAV)

**S1 File.**
(DOCX)

## Acknowledgments

The authors are grateful to the students for participating in the study.

## Author Contributions

**Conceptualization:** Edmund Ndudi Ossai, Lawrence Ulu Ogbonnaya.

**Data curation:** Irene Ifeyinwa Eze, Chukwuma David Umeokonkwo, Chukwuemeka Obioma Izuagba.

**Formal analysis:** Edmund Ndudi Ossai.

**Methodology:** Edmund Ndudi Ossai, Irene Ifeyinwa Eze, Chukwuma David Umeokonkwo, Chukwuemeka Obioma Izuagba, Lawrence Ulu Ogbonnaya.

**Supervision:** Chukwuma David Umeokonkwo, Lawrence Ulu Ogbonnaya.

**Writing – original draft:** Irene Ifeyinwa Eze.

**Writing – review & editing:** Edmund Ndudi Ossai, Irene Ifeyinwa Eze, Chukwuma David Umeokonkwo, Chukwuemeka Obioma Izuagba, Lawrence Ulu Ogbonnaya.

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
