## [Decision Letter · Decision Letter 0]

2 Mar 2023

PONE-D-22-18325Readiness, barriers, and attitude of students towards online medical education amidst COVID-19 pandemic: a study among medical students of Ebonyi State University Abakaliki, NigeriaPLOS ONE

Dear Dr. Eze,

Thank you for submitting your manuscript to PLOS ONE. After careful consideration, we feel that it has merit but does not fully meet PLOS ONE’s publication criteria as it currently stands. Therefore, we invite you to submit a revised version of the manuscript that addresses the points raised during the review process.

We look forward to receiving your revised manuscript.

Kind regards,

Yaser Mohammed Al-Worafi

Academic Editor

PLOS ONE

Journal Requirements:

Reviewers' comments:

Reviewer's Responses to Questions

**Comments to the Author**

1. Is the manuscript technically sound, and do the data support the conclusions?

Reviewer #1: Partly

Reviewer #2: Yes

2. Has the statistical analysis been performed appropriately and rigorously? 

Reviewer #1: Yes

Reviewer #2: Yes

3. Have the authors made all data underlying the findings in their manuscript fully available?

Reviewer #1: Yes

Reviewer #2: Yes

4. Is the manuscript presented in an intelligible fashion and written in standard English?

Reviewer #1: Yes

Reviewer #2: Yes

5. Review Comments to the Author

Reviewer #1: This study describes the barriers to online learning during COVID-19 in Nigeria which is a gap in literature. However there are some changes that need to be made and the recommendations are as follows:

Line 130,131 --Reword the sentence for clarity "The university was eventually re-opened by February 2021. The study occurred between September and October 2021, about seven months after the school was re-opened"

Please detail the questionnaire used in the study by referencing or adding it in the appendix

Table 1,2,3 - merge the frequency and percentage in one column

Table 2- merge the frequency and percentage in one column, change "i phone" to "iphone", change "Internet modern to internet modem. The email address, webinar and internet access question in the table without a Yes should be structured.

Reviewer #2: Dear

I realize that authors have many journals to consider when they want to publish their work, so I appreciate your interest in PLOS ONE; I am very sorry not to be able to write in a more positive way. It is evident that you have put a great deal of effort into this project and I want to praise your efforts. Fortunately, the actual contribution from your study is clear and, the manuscript as currently written suggests that it might be suitable for sharing information about online medical education amidst COVID-19 pandemic topic, but the paper that you reported, needs few major edits. I should like to thank you for give me an opportunity to consider this work for publication.

It may be that the you would like to consider resubmitting it, in which case I hope that the comments from my review may help you to revise it before resubmitting it. These comments are given below.

Best Regards

- General review:

The paper is very interesting, but essentially needs to be rewritten to make it more linear scientifically and narrative structured

- Introduction section:

is too poor; many references are missing in the many sentences; In scientific literature is not possible to have the parts of paper without references;

better describe the context, the reasons for this study and the need to implement the topic of online medical education;

- Material and Methods:

Please indicate which Statement of reporting was used to construct and conduct the study, e.g. in according to the Checklist for Reporting Results of Internet E-Surveys (CHERRIES) guidelines (Eysenbach, 2004) and or STrengthening the Reporting of Observational Studies in Epidemiology (STROBE) (von Elm et al., 2007);

Define whether questionnaire development and pre-testing questionnaire processes were used;

- Results:

insert a more descriptive part and greater analysis of the results in a reasoned and narrative way, not just describing the tables.

- in discussion:

Discussions should be reviewed in light of the overall improvement of the paper. Redundant sentences and prewritten information should be avoided. Focus on take-home messages and how that information impacts in the online healthcare education and how its implementations will be exploited in the near future.

introduce a section of strengths also;

- Tables:

could be improved in style reporting.

- Reference:

is very poor. I suggest to add the follow references in the introduction section:

Online teaching in physiotherapy education during COVID-19 pandemic in Italy: a retrospective case-control study on students' satisfaction and performance.

Rossettini G, Geri T, Turolla A, Viceconti A, Scumà C, Mirandola M, Dell'Isola A, Gianola S, Maselli F, Palese A.

BMC Med Educ. 2021 Aug 30;21(1):456. doi: 10.1186/s12909-021-02896-1.

Digital Entry-Level Education in Physiotherapy: a Commentary to Inform Post-COVID-19 Future Directions.

Rossettini G, Turolla A, Gudjonsdottir B, Kapreli E, Salchinger B, Verheyden G, Palese A, Dell'Isola A, de Caro JX.

Med Sci Educ. 2021 Nov 4;31(6):2071-2083. doi: 10.1007/s40670-021-01439-z. eCollection 2021 Dec.

Minor Edits:

- explode the acronym ITC

6. PLOS authors have the option to publish the peer review history of their article (what does this mean?). If published, this will include your full peer review and any attached files.

Reviewer #1: No

Reviewer #2: No

---

## [Author Response · Author response to Decision Letter 0]

16 Mar 2023

Dear Editor,

 The dataset has been uploaded under the supplementary file.

Dear Reviewers, 

Thank you for the comment, input, and recommendations; they have helped to enrich the paper. Below are the responses to the comments.

Reviewer #1: 

This study describes the barriers to online learning during COVID-19 in Nigeria which is a gap in literature. However there are some changes that need to be made and the recommendations are as follows:

Response: Thank you for the observations and input. The corrections have been made, and the recommendations have been acted on.

Line 130,131 --Reword the sentence for clarity "The university was eventually re-opened by February 2021. The study occurred between September and October 2021, about seven months after the school was re-opened"

Response: Thank you. The statement has been rephrased for clarity; Lines 136-137

Please detail the questionnaire used in the study by referencing or adding it in the appendix

Response: The questionnaire used has been included as a supporting document. Thank you.

Table 1,2,3 - merge the frequency and percentage in one column

Response: Thank you for the input. The frequency and percentage have been merged into one column. 

Table 2- merge the frequency and percentage in one column, change "i phone" to "iphone", change "Internet modern to internet modem. The email address, webinar and internet access question in the table without a Yes should be structured.

Response: Thank you for the input. The observed corrections in table 2 have been effected; lines 196

Reviewer #2: 

I realize that authors have many journals to consider when they want to publish their work, so I appreciate your interest in PLOS ONE; I am very sorry not to be able to write in a more positive way. It is evident that you have put a great deal of effort into this project and I want to praise your efforts. Fortunately, the actual contribution from your study is clear and, the manuscript as currently written suggests that it might be suitable for sharing information about online medical education amidst COVID-19 pandemic topic, but the paper that you reported, needs few major edits. I should like to thank you for give me an opportunity to consider this work for publication.

It may be that the you would like to consider resubmitting it, in which case I hope that the comments from my review may help you to revise it before resubmitting it. These comments are given below.

Best Regards

- General review:

The paper is very interesting, but essentially needs to be rewritten to make it more linear scientifically and narrative structured

Response: Thank you for the observations and corrections. The paper has been revised to make it more scientific and structured. 

- Introduction section:

is too poor; many references are missing in the many sentences; In scientific literature is not possible to have the parts of paper without references;

better describe the context, the reasons for this study and the need to implement the topic of online medical education;

Response: Thank you for the valuable input, which enriched the paper. The introduction has been revised accordingly, Lines 67-116

- Material and Methods:

Please indicate which Statement of reporting was used to construct and conduct the study, e.g. in according to the Checklist for Reporting Results of Internet E-Surveys (CHERRIES) guidelines (Eysenbach, 2004) and or STrengthening the Reporting of Observational Studies in Epidemiology (STROBE) (von Elm et al., 2007); 

Response: Thank you for the input. The Strengthening the Reporting of Observational Studies in Epidemiology (STROBE) was used to construct and report the study; lines 140-141

Define whether questionnaire development and pre-testing questionnaire processes were used;

Response: Yes, as indicated in the manuscript, the questionnaire was designed by the researchers after reviewing some literature and pretested among medical students in another university. The statement has been more clearly defined. Lines 147-150

- Results:

insert a more descriptive part and greater analysis of the results in a reasoned and narrative way, not just describing the tables.

Response: Thank you for the input. The results have been revised for a better presentation; lines 177-231

- in discussion:

Discussions should be reviewed in light of the overall improvement of the paper. Redundant sentences and prewritten information should be avoided. Focus on take-home messages and how that information impacts in the online healthcare education and how its implementations will be exploited in the near future.

Response: Thank you for your contribution. The discussion has been modified for better flow; lines 234-296

introduce a section of strengths also;

Response: The strengths of the study have been added; lines 286-288. Thank you

- Tables:

could be improved in style reporting.

Response: The tables have been modified and formatted. Thank you

- Reference:

is very poor. I suggest to add the follow references in the introduction section:

Online teaching in physiotherapy education during COVID-19 pandemic in Italy: a retrospective case-control study on students' satisfaction and performance.

Rossettini G, Geri T, Turolla A, Viceconti A, Scumà C, Mirandola M, Dell'Isola A, Gianola S, Maselli F, Palese A.

BMC Med Educ. 2021 Aug 30;21(1):456. doi: 10.1186/s12909-021-02896-1.

Digital Entry-Level Education in Physiotherapy: a Commentary to Inform Post-COVID-19 Future Directions.

Rossettini G, Turolla A, Gudjonsdottir B, Kapreli E, Salchinger B, Verheyden G, Palese A, Dell'Isola A, de Caro JX.

Med Sci Educ. 2021 Nov 4;31(6):2071-2083. doi: 10.1007/s40670-021-01439-z. eCollection 2021 Dec.

Response: Thank you for the input. The references have been reviewed, and necessary additions and corrections made; lines 332-402 

Minor Edits:

- explode the acronym ITC

Response: The acronym ICT, has been corrected; line 230

---

## [Editor Report · Decision Letter 1]

13 Apr 2023

Readiness, barriers, and attitude of students towards online medical education amidst COVID-19 pandemic: a study among medical students of Ebonyi State University Abakaliki, Nigeria

PONE-D-22-18325R1

Dear Dr. Irene, 

We’re pleased to inform you that your manuscript has been judged scientifically suitable for publication and will be formally accepted for publication once it meets all outstanding technical requirements.

Kind regards,

Yaser Mohammed Al-Worafi

Academic Editor

PLOS ONE
---

## [Editor Report · Acceptance letter]

17 Apr 2023

PONE-D-22-18325R1 

Readiness, barriers, and attitude of students towards online medical education amidst COVID-19 pandemic: a study among medical students of Ebonyi State University Abakaliki, Nigeria 

Dear Dr. Eze:

I'm pleased to inform you that your manuscript has been deemed suitable for publication in PLOS ONE. Congratulations! Your manuscript is now with our production department. 

Kind regards, 

on behalf of

Professor Yaser Mohammed Al-Worafi 

Academic Editor

PLOS ONE